# Dynamics of Dynamin-Related Protein 1 in Alzheimer’s Disease and Other Neurodegenerative Diseases

**DOI:** 10.3390/cells8090961

**Published:** 2019-08-23

**Authors:** Darryll Oliver, P. Hemachandra Reddy

**Affiliations:** 1Department of Internal Medicine, Texas Tech University Health Sciences Center, Lubbock, TX 79430, USA; 2Garrison Institute on Aging, Texas Tech University Health Sciences Center, Lubbock, TX 79430, USA; 3Garrison Institute on Aging, South West Campus, Texas Tech University Health Sciences Center, Lubbock, TX 79413, USA; 4Department of Cell Biology and Biochemistry, Texas Tech University Health Sciences Center, Lubbock, TX 79430, USA; 5Department of Pharmacology and Neuroscience, Texas Tech University Health Sciences Center, Lubbock, TX 79430, USA; 6Department of Neurology, Texas Tech University Health Sciences Center, Lubbock, TX 79430, USA; 7Department of Speech, Language and Hearing Sciences, Texas Tech University Health Sciences Center, Lubbock, TX 79430, USA

**Keywords:** mitochondria, dynamin-related protein 1, Alzheimer’s disease, aging, Huntington’s disease, Parkinson’s disease, diabetes, obesity, mitochondrial division inhibitor

## Abstract

The purpose of this article is to highlight the role of dynamin-related protein 1 (Drp1) in abnormal mitochondrial dynamics, mitochondrial fragmentation, autophagy/mitophagy, and neuronal damage in Alzheimer’s disease (AD) and other neurological diseases, including Parkinson’s, Huntington’s, amyotrophic lateral sclerosis, multiple sclerosis, diabetes, and obesity. Dynamin-related protein 1 is one of the evolutionarily highly conserved large family of GTPase proteins. Drp1 is critical for mitochondrial division, size, shape, and distribution throughout the neuron, from cell body to axons, dendrites, and nerve terminals. Several decades of intense research from several groups revealed that Drp1 is enriched at neuronal terminals and involved in synapse formation and synaptic sprouting. Different phosphorylated forms of Drp1 acts as both increased fragmentation and/or increased fusion of mitochondria. Increased levels of Drp1 were found in diseased states and caused excessive fragmentation of mitochondria, leading to mitochondrial dysfunction and neuronal damage. In the last two decades, several Drp1 inhibitors have been developed, including Mdivi-1, Dynasore, P110, and DDQ and their beneficial effects tested using cell cultures and mouse models of neurodegenerative diseases. Recent research using genetic crossing studies revealed that a partial reduction of Drp1 is protective against mutant protein(s)-induced mitochondrial and synaptic toxicities. Based on findings from cell cultures, mouse models and postmortem brains of AD and other neurodegenerative disease, we cautiously conclude that reduced Drp1 is a promising therapeutic target for AD and other neurological diseases.

## 1. Introduction

Mitochondrial dynamics consists of mitochondrial motility, fission, fusion, and mitophagy [1]. Mitochondrial motility via microtubules allows mitochondria to traverse the cell, and maintain adequate distribution, which is especially important in neuronal cells with extensive axons and numerous dendrites [1,2,3]. Fission permits mitochondrial reproduction, very similar to binary fission in bacteria, and further their propagation throughout the cell [4]. Fission also is used to regulate apoptosis, mitophagy, and changes in bioenergetics demands. Mitochondrial fusion is a means by which mitochondria merge into one, coalescing membranes and cellular contents, including mitochondrial DNA (mtDNA) to make a larger and more resource rich organelle [4]. 

Mitophagy is a mitochondria specific autophagy meant to excise damaged and dysfunctional mitochondria and their contents from the cell [5]. The end result is a healthier network of mitochondria, with a reduction in reactive oxygen species (ROS) production, and damaged mtDNA, along with an increased overall biogenesis to ROS production ratio [1,4,5,6].

Mitochondrial dynamics have multiple functions for cells and tissues. Ca^2+^ regulation is necessary for ion electrolyte balance to maintain membrane potential, Ca^2+^ use as a second messenger in G-protein regulation, and sarcomere Ca^2+^ supplication for muscle flexion and extension [7,8,9]. ROS production is used to trigger autophagy within cells, including mitochondria specific mitophagy [4]. Alzheimer’s disease (AD) related loss of mitochondrial membrane integrity, triggered by loss of cardiolipin, loss of potential, and/or oxidative damage from excessive ROS production, leads to release of cytochrome *c*, which triggers caspase-9 initiated apoptotic pathway of the cell [10]. This pathway is integral for regulation of the cell cycle, and especially affects ratio of cellular growth, division, senescence, and death [11]. The size and distribution of mitochondria throughout the cell also determines the magnitude of ATP available for biochemical reactions. This ATP availability then affects (1) rates of transcription and translation, (2) ability to respond to damage, (3) cellular diversity, including pluripotent cell distribution and tissue development, (4) immune response, and (5) resistance to diseases, including neurological diseases, cancer, mutagenesis and metastasis, muscular dystrophy, ischemic heart disease, chronic liver disease, and liver cirrhosis [1,2,4,10,12,13,14,15,16,17,18,19,20,21,22,23]

The purpose of this article is to highlight the role of dynamin-related protein 1 (Drp1) in abnormal mitochondrial dynamics in AD and other neurodegenerative diseases. This article also summarizes the recent developments in phosphorylated Drp1, current status of Drp1 inhibitors, and how reduced Drp1 protects against excessive fragmentation of mitochondria in Alzheimer’s and other neurodegenerative diseases.

## 2. Mitochondrial Fusion and Fission

Mitochondrial fusion and fission have a dynamic relationship, with an ebb and flow between the expression of fusion and fission proteins maintaining a healthy mitochondrial network within the cell, determined by size, distribution, and biogenesis [4] (Figure 1). The main proteins are Drp1, Mitochondrial fission factor (MFF), Fission-1 (Fis1), and the homologues Mid49 and Mid51 for the fission pathway; mitofusins 1 (Mfn1) and 2 (Mfn2), and optic atrophy 1 (OPA1) are proteins that usher mitochondria along the fusion pathway [4,24].

During fusion, Mfn1 and Mfn2 control the fusion of the outer mitochondrial membrane (OMM), while OPA1 is localized to the inner mitochondrial membrane (IMM) [24]. OPA1 is also responsible for mtDNA maintenance as well as structural integrity of IMM cristae [24]. The two mitochondria first join at their OMM, and Mfn1 and 2 work to bind the OMM, after which OPA1 works to bind the IMM [24]. The fusion of two mitochondria increases the mtDNA copy number, and rejuvenates components within the organelle [6].

Genetic material is replicated prior to the fission of a mitochondrion [25]. Fission proteins Fis1, Mff, Mid49, and Mid51 localize to the target area of division on the mitochondrion, and recruit Drp1 to target site [24]. Drp1 and the associated division proteins Fis1, Mff, Mid49, and Mid51 create tension at the target location for division, which causes a “pinch” in the mitochondrial membrane [24]. The depression in the “pinch” increases until the membranes on either side of the “pinched” area fuse with each other, causing the mitochondrion to become two mitochondria [24].

## 3. Dynamin-Related Protein 1

Mitochondrial Drp1 protein is integral for maintaining balance in mitochondrial dynamics, regulating fission, fusion, mitophagy, and even motility. Primarily, it serves as a mitochondrial fission factor, inducing mitochondrial division, but when downregulated, indirectly promotes fusion [26]. Drp1 regulated fission may be used to excise damaged portions of mitochondria for mitophagy, and reduction in Drp1 recruits parkin for mitophagy [27]. Excessive fission or fusion also impedes mitochondrial transport [28,29]. Thus, downregulation or upregulation of its transcription can be used to measure incremental changes in mitochondrial dynamics. Drp1 is also necessary for regulating lifespan. Rana et al. (2017) augmented Drp1 expression in *Drosophila* midlife flies, inducing greater amount of mitochondrial fission, and successfully extended their lifespan [30].

## 4. Drp1 Structure

The Drp1 protein was first reported by Shin et al. (1997) [31], and is transcribed from the 12p11.21 gene and 11q23 gene in humans and mice/rats respectively [24]. Drp1 is also known as DVLP, DLP1, Hdyn IV, and Dymple and may be found throughout the cell, including its mitochondria, cytoplasm, various vesicles including Golgi apparatus and peroxisomes [24].

The protein itself is made of 736 residues, folded and organized into a quaternary structure with four domains to function as a dynamin GTPase protein [24]. In humans, six isoforms of the human protein are produced by alternative splicing. Isoform 1 is the brain-specific, longer isoform. Isoform 2 and isoform 3 are predominantly expressed in testis and skeletal muscles respectively. Isoform 4 is weakly expressed in brain, heart, and kidney. Isoform 5 is dominantly expressed in liver, heart, and kidney. Isoform 6 is expressed in neurons. The largest one consists of 736 amino acids with a calculated molecular mass of 81.6 kDa; in variant 2, exon 15 is spliced out and has a calculated molecular mass of 710 amino acids; in variant 3, exons 15 and 16 are spliced out and have a total of 699 amino acids; variant 4 has 725 amino acids; variant 5, 710 amino acids; and variant 6, 749 amino acids [32].

Similar to human Drp1, multiple variants of Drp1 have been found in the mouse. In the mouse, variant 1 consists of 712 amino acids, with a calculated molecular mass of 78.3 kDa; in variant 2, exon 3 is spliced out, and in variant 3, exons 15 and 16 are spliced out. It is possible that multiple splice variants are present in mouse Drp1 [32].

The presence of a highly conserved GTPase domain in Drp1 indicates that Drp1 is involved in essential functions of mitochondria and cell. The dynamin proteins have a large GTPase domain consisting of ~300 residues [26]. The four domains are the N-terminal GTPase domain, middle domain, variable or B domain, and the C-terminal GTPase effector domain [24] (Figure 2). 

For Drp1 GTPase activity, the primary messenger molecule binds the target receptor causing a conformational change, activating the receptor [26]. This conformational change allows it to bind an inactive protein. Guanosine Diphosphate (GDP) then dissociates from the GTPase domain, while Guanosine Triphosphate (GTP) associates to the protein active site activating the Drp1 [26]. The activated protein disengages the receptor, and the protein then engages the target effector protein initiating the cellular response [26]. Thus, the hydrolysis of GTP here results in conformational change in Drp1 necessary for its activity [26].

The Drp1 protein is translated in the cytosol, where it exists in a dimer-tetramer-equilibrium, and is directed to the OMM where it operates its key role in mitochondrial fission [26]. When directed to the OMM, its structure is augmented to form oligomers, which amalgamate at the target for scission [26]. For Drp1 specifically, GTP binding is necessary for induction of Drp1 assemblage [26]. Drp1 also helps to disassemble peroxisomes [26].

There are various types of post-translation modifications for Drp1 that may affect rate of mitochondrial fission. These include phosphorylation, SUMOylation (or sumoylation), ubiquitination, and S-Nitrosylation. An example of Drp1 phosphorylation is at the Ser-616 site by Cdk1/cyclin B, which does not directly alter GTPase activity, but likely indirectly induces fission due to Drp1activity with additional fission factors. Phosphorylation at Ser-585 and 637, also induce fission. When the small ubiquitin-like modifier (SUMO) protein attaches to Drp1 it assists with Drp1 modification, particularly at the OMM. The result of sumoylation of Drp1 however is not fully understood. Ubiquitin protein interaction normally signals for elimination of a cellular element, such as with various types of autophagy within the cell. With Drp1 and Fis1, the “RING-CH E3 (MARCH-V, MARCH5, MITOL) ubiquitin ligase” of the OMM ubiquitinates the proteins, resulting in an increase in mitochondrial fission. S-Nitrosylation of Drp1 alters the protein conformation, augmenting GTPase activity, upregulating fission. In fact, S-Nitrosylation at Cys-644, induced from interaction with amyloid-beta in AD, causes excessive fission and neurotoxicity [23,24,26].

## 5. Drp1 Functions

Dynamin-related protein 1 is multifactorial protein that performs several critical and important functions of the mitochondria and cell [32,33]. As mentioned above, Drp1 is largely involved in mitochondrial division and distribution, peroxisomal fragmentation, SUMOylation, phosphorylation and cell death, ubiquitination, and synapse formation and synapse sprouting [32,33].

### Drp1 Knockout Studies

Mouse double knockouts of Drp1 are embryonic lethal, and embryos cannot survive beyond 11–12 days [34,35] indicating that Drp1 is necessary for cell survival. However, heterozygote Drp1 knockout (Drp1+/−) mice are normal in terms of lifespan, fertility, and viability; and phenotypically, the Drp1+/− mice are not different from WT mice. To determine the effects of a partial reduction of Drp1 in the WT mice, we recently compared synaptic, dendritic, and mitochondrial proteins in Drp1+/− and WT mice [36]. The Drp1+/− mice showed enhanced synaptic and mitochondrial fusion proteins relative to the WT mice, and the Drp1+/− mice showed significantly reduced free radicals and lipid peroxidation levels compared to the WT mice. These findings suggested that a partial reduction in Drp1 is beneficial.

## 6. Drp1 and Abnormal Mitochondrial Dynamics

Mitochondrial dysfunction rooted in Drp1 imbalance occurs in pathology of various diseases including Alzheimer’s, Down syndrome, multiple sclerosis, amyotrophic lateral sclerosis (ALS), Parkinson’s, and triple repeat diseases such as Huntington’s disease, Spinobulbar muscular atrophy, spinocerebellar atrophy 1, and others [32,37,38,39,40,41,42]. Increased mitochondrial fission and reduced fusion are prominent features in aging, AD, and a large number of neurological diseases [32,37,38,39,40,41,42]. The following is a summary of neurodegenerative dysfunction caused by Drp1 hyperactivity inducing mitochondrial dysfunction.

### 6.1. Drp1 and Alzheimer’s Disease

With the understanding that excessive mitochondrial fission was a major component of AD pathology, researchers Baek et al. [43] set out to determine whether a reduction in fission by inhibiting Drp1 would alleviate AD pathology. These researchers utilized hemizygous double-transgenic APP/PS1 mice for their experiment, and a group of age matched wild-type mice for their control; mice were either treated or untreated with mitochondrial division inhibitor 1 (Mdivi-1), an indirect Drp1 inhibitor, at either 10 mg/kg per day, or 40 mg/kg per day. The experiment consisted of 34 days of physical exercise, with tests on day 35 for memory retention, passive avoidance on day 37, and then killings on day 40 for brain sample harvesting. The mice untreated with Mdivi-1 were noted to have stunted, round mitochondria, with inhibited anterograde mitochondrial transport. Furthermore, there was a higher rate of reactive oxygen species production, including superoxide, and reduced biogenesis, which is likely what caused synaptic damage. Diseased mice demonstrated reductions at the synapse, greater neuron loss, as well as meager dendritic growth [43].

In 2017, Wang and colleagues also studied the effect of Mdivi-1 on mitochondrial dynamics, particularly fission in CRND8 mice (transgenic APP strain) [44]. In Mdivi-1 treated and untreated 3-months-old CRND8 mice, they characterized mitochondrial fragmentation, distribution, and mitochondrial function in the pyramidal neurons [44]. They found Mdivi-1 treatment rescued both mitochondrial fragmentation and distribution deficits and improve mitochondrial function in the CRND8 neurons both in vitro and in vivo. The mitochondrial dynamic deficits, Aβ1-42/Aβ1-40 ratio and amyloid deposition were reduced by mdivi-1 treatment. Further, cognitive deficits were prevented in Mdivi-1 treated CRND8 mice [44].

Reddy et al. [45] experienced similar results when measuring Drp1 over expression in AD mouse neuroblastoma (N2a) cells. Cell lines were grown to express mutant Amyloid Precursor Protein (mAPP) toxicity, treated with Aβ_42_ after 3 days, and treated with Mdivi1 for 24 h, or left untreated [45]. There were fragmentations in mitochondria of the N2a cells with Aβ toxicity, including small, spherical size, and excessive fragmentation [45].

### 6.2. Drp1 and Huntington’s Disease

Huntington’s disease (HD) is a neurodegenerative disease characterized by loss of motor control marked involuntary movement, cognitive decline, loss of body weight, and irritable mood affecting adults 30–40 years, and worsening with age [46]. In HD just as in AD, increased levels of Drp1 are a cause of mitochondrial dysfunction [46]. The mutant huntingtin protein (mHtt), an aberrant version of the huntingtin (Htt) protein with excessive repeats of glutamine residues (over 36 repeats), has been noted to affect Drp1 GTPase activity, increasing mitochondrial fission [3,47] (Figure 3).

Song et al. (2011) [47] and Shirendeb et al. (2012) [3] noted the mHtt interaction with Drp1 GTPase domain, propagating enzyme activity leading to disproportionate mitochondrial fission [3,47]. Song et al. assessed rat cortical neurons and human HD fibroblasts, expressing mHtt, with mice expressing full length human Htt, with either 18Q or 128Q repeats, and noted changes over time [47]. It was found that mitochondrial anterograde transport down the axon decreased in velocity and frequency, inhibiting synapse; mitochondrial fission increased to the point of fragmentation, producing stunted and round mitochondria, rather than full, oblong shaped organelles; autophagy increased, and mitochondrial numbers decreased; in addition, mitochondrial dysfunction increased, evident by the increase of ROS, measured by H_2_O_2_ concentration [47].

In the Shirendeb et al. [3] study, experimental mice were transfected with full-length human Htt gene with 97 CAA and CAG (mixed) repeats, to express HD using bacterial artificial chromosome (BAC). Analyses were made after growing mice for at least 2 months (when mice started to show motor deficits) and sacrificed for post mortem autopsy. Shirendeb et al. [3] noted that there was mitochondrial fragmentation, as well as inhibition of axonal transport of mitochondria towards the synapse, which also reduces the number of vesicles exchanged within the synapse. In addition, neurons transfected with the HD mutation were found to have deformed dendrites, of meager sizing [3].

### 6.3. Drp1 and Amyotrophic Lateral Sclerosis

Amyotrophic Lateral Sclerosis (ALS) or Lou Gehrig’s disease is a motor neuron disease, affecting the upper and lower motor neurons of the spinal cord [48,49]. Disease progressively worsens from weakness, stiffness, and soreness in muscles, to paralysis [48,49]. Mutation of the superoxide dismutase 1 (SOD1) gene is thought to be distributed sporadically throughout the human population as the cause is unknown, the effect of which is the pathogenesis of ALS [48,49]. A study by Tafuri et al. indicated that mitochondrial dysfunction, particularly reduction in IMM folding, fragmentation, excessive ROS production, and stunted, spherical mitochondria was connected to mutant SOD1 [48]. The main factor being mutant SOD1 localization to the inter membrane space, where much of the ROS produced accumulates [48]. In ALS, just as with AD and HD, upregulation of Drp1 activity, as well as Fis1 expression, were at the center of mitochondrial fragmentation [49]. In an experiment by Joshi et al., fibroblasts with mutant SOD1 were analyzed against control patients, and those expressing the mutant SOD1 protein had considerably reduced mitochondrial membrane potential; greater ROS production (upwards of 186% and 250% greater than the control); Drp1 hyperactivity via phosphorylation at Ser-616 by CDK 1/cyclin B or CDK5 and decrease in phosphorylation at Ser-637 by calcineurin, measured via Western blot analysis (both increasing mitochondrial fission); greater Drp1 and Fis1 interaction measured via co-immunoprecipitation; greater levels of p62 attracted to mitochondria; increased mitochondrial fragmentation, and disproportioned autophagy [49].

### 6.4. Drp1 and Parkinson’s Disease

Parkinson’s disease (PD) is only surpassed by AD as the most common form of neurodegenerative disease [6]. The source of its pathology is the death of dopaminergic neurons (induced by dopamine, a neurotransmitter), due to gene mutations at approximately 20 loci. The damage is expressed as dysfunction in the motor cortex, when there is loss of dopaminergic signaling from the substantia nigra, and affects motor and non-motor neurons [6]. PD patients may experience loss of motor control including postural instability, slowed movements or “bradykinesia,” and resting tremors [6]. Additional symptoms include depression, hallucinations, dementia, REM sleep disorders autonomic dysfunction and olfactory impairments [6]. Similar to AD, there is a soluble protein precipitated in the central nervous system of PD patients called α-synuclein (α-Syn) that amalgamates intracellularly into an insoluble oligomer called a Lewy body, the toxicity of which is not fully understood regarding its induction of PD pathology, whether it is a source of disease or a mere consequence [6]. It is known that α-Syn affects mitochondrial dynamics, and directly or indirectly impacts Drp1 activity. For instance, leucine-rich repeat kinase 2 (LRRK2) plays a role of activating Drp1 by transferring phosphate to increase mitophagy, may experience a mutation as a result of PD inducing instead fission and further, fragmentation [6,28].

With mitophagy, PTEN-induced putative kinase 1 (PINK1) serves the role of initiating the mitophagy pathway [1]. PINK1 changes its conformation and is activated once the mitochondrion loses its membrane potential, and shortly thereafter recruits parkin to the OMM to initiate ubiquitination, recruiting additional mitophagy proteins [1]. PINK1 also serves the role of phosphorylation of the Rho GTPase domain of Mitochondrial Rho protein (MIRO1—Ca^2+^ sensor which regulates mitochondrial anterograde transport), to inhibit the motility of damaged mitochondria set for mitophagy [1,50]. Conversely, PD pathology prevents PINK1 from inhibiting MIRO1, and effectively causes mitophagy dysfunction [50]. With damaged mitochondria, PINK1 also serves a role of activating Drp1 for scission necessary for mitophagy, by inducing the dissociation of protein kinase A (PKA) from A-kinase Anchoring Protein (AKAP1), allowing the dephosphorylation of Drp1 at Ser-637 (note: normally AKAP-1 bound PKA inhibits Drp1 via phosphorylation at Ser-637 to reduce fission) [29]. However, with PD there is a loss of PINK1, and thus AKAP1 continues to be inhibited by PKA, inhibiting the Drp1 induced scission necessary for successful mitophagy [51].

### 6.5. Drp1 and Multiple Sclerosis

Multiple Sclerosis (MS) is a sporadic neurodegenerative disease of the CNS, where the body attacks itself, particularly the oligodendrocytes that form the myelin sheath insulation of nerve fibers, causing the nerves to eventually degrade [52]. The effects of the disease vary from loss of motor control, weakness in limbs, impaired vision, and tiredness [52]. In MS patients, the inflammation which induces this attack on oligodendrocytes is initiated by a cytokine known as tumor necrosis factor (TNF), and these oligodendrocytes have shown to demonstrate hyperactivity of Drp1, and excessive mitochondrial fragmentation [53].

### 6.6. Drp1 and Down Syndrome

Down syndrome (DS) results from an additional copy of chromosome 21, in humans, and is demonstrated by cognitive disabilities and hypotonia (meager muscle tone) [54]. Mitochondrial dysfunction is associated with Down syndrome as well, and upregulation of Drp1, causing excessive fragmentation of mitochondria that is associated with abnormal differentiation of neural progenitor cells in DS brains [54].

### 6.7. Drp1 and Diabetes/Obesity

Increased Drp1 levels and abnormal mitochondrial dynamics were reported in diabetes and obesity conditions [55,56,57,58,59]. Recently, Huang and colleagues (2015) [58] extensively studied Drp1 and GSK3 beta levels in SK cell cultures and db/db mouse models of diabetes. Using a mouse model of type 2 diabetes (*db/db* mice) and a human neuronal cell line treated with high concentration of glucose, they found aberrant increased Drp1 and GSK 3β levels altered mitochondrial morphology, reduced ATP production, and impaired activity of complex I. These mitochondrial abnormalities are induced by imbalanced mitochondrial fusion and fission via a GSK3β/Drp1-dependent mechanism. Modulation of the Drp1 pathway or inhibition of GSK3β activity restores hippocampal long-term potentiation that is impaired in *db/db* mice [58].

Makino et al. 2010 [59] studied mitochondrial dynamics and mitochondrial function in mouse coronary endothelial cells (MCECs) isolated from diabetic mice. They found increased levels of Drp1 and mitochondrial fragmentation. Elongated mitochondrial tubules were observed in MCECs isolated from control mice, whereas mitochondria in MCECs from diabetic mice exhibited augmented fragmentation. The level of fusion protein Opa1 was significantly decreased. Diabetic MCECs exhibited significantly higher O_2-_ concentrations in cytosol and mitochondria than control MCECs. Administration of the O_2-_ scavenger TEMPOL to diabetic mice for 4 weeks led to a significant decrease in mitochondrial fragmentation without altering the levels of Opa1 and Drp1 proteins in MCECs. High-glucose treatment for 24 h significantly induced mitochondrial fragmentation, which was restored by TEMPOL treatment. In addition, excess O_2-_ production, either in cytosol or in mitochondria, significantly increased mitochondrial fragmentation [59].

## 7. Mechanism of Mitochondrial Dysfunction via Drp1 Phosphorylation

Amyloid beta and phosphorylated tau interactions with mitochondria causes upregulation of Drp1 in AD [40,60]. One dysfunctional mechanism that causes excessive fission, is phosphorylation of Drp1 due to Aβ interaction. As shown in Table 1, Drp1 has been reported to be phosphorylated at three main sites, Ser-616 (aka Ser-579), Ser-637, and at Thr-595 by LRRK2 [28] (though phosphorylation may also occur at Ser-40, Ser-44, Ser-585, Ser-656, and Ser-696 as well). Phosphorylation propagated by glycogen synthase kinase 3β (GSK3β) induces activity of the Drp1 GTPase domain, causing mitochondrial fission. Furthermore, Aβ interaction with Cyclin dependent kinase 5 (CDK5) induces phosphorylation of Drp1 at residue Ser-579, which leads cysteine-aspartic acid protease 3 (caspase 3) to be cleaved, executing indiscriminate cell apoptosis [61]; the result being AD pathology of neuronal death for example. Aβ mediated activation of CDK5 can cause phosphorylation of Ser-585 which upregulates Drp1, which inhibits biogenesis, increases oxidative damage caused by ROS overproduction, and causes increase of glutamate binding to *N*-methyl-D-Aspartate (NMDA) receptors, inhibiting Ca^2+^ cation flow necessary for memory and learning, the cognitive decline characteristic of AD pathology.

However, there are caveats to consider with Drp1 inhibition, including its requirement for healthy mitochondrial dynamics. As mentioned before, it is regularly upregulated and downregulated throughout a healthy mitochondrial life cycle. Furthermore, phosphorylation of Drp1 may be used to increase fission, and also decrease fission. For example, phosphorylation at Ser-616 by CDK 1/cyclin B or CDK5 increases mitochondrial fission, but phosphorylation at Ser-637 by calcineurin reduces mitochondrial fission. Thus, a drug developed to treat excessive fission would need to be designed for specific sites of phosphorylation or dephosphorylation. 

Roy et al. (2015) [68] did breed transgenic MyH6-Cre mice, deficient in capability of expressing Drp1 gene, and the effects were disastrously debilitating. These included bradycardia, and decreased stroke volume, leading to death between postnatal days 9 and 11. Mitochondrial dynamic dysfunction led to cardiac arrest, yet mitochondrial dynamics are just as important to neuronal function, and these mice did not live long enough to note the likely damaging effects to motor control. The double knockout of Drp1 led to elongated, interconnected mitochondria, dysfunctional due to oxidative damage, incomplete mitophagy dysfunction where mitochondria were unable to colocalize autophagosomes with lysosomes, and inability to complete fission, which further disrupted mitophagy and mitochondrial transport. Histological activity assays conducted by Roy et al. (2015) demonstrated reduced biogenesis in cardiomyocytes, due to inhibition of electron transport chain complex activity [68].

In Huntington’s disease, induction of mitogen-activated protein kinase 1 (MAPK1) phosphorylates Ser-616, which upregulates Drp1 activity causing mitochondrial fragmentation, depolarization, and ROS production [69]. In ischemic mice, mitochondrial A-kinase anchoring protein 1 (AKAp1) numbers are reduced, which causes propagation of ROS damage, mitochondrial fragmentation, and Ca^2+^ ion potential imbalance, due to de-phosphorylation of Drp1 at Ser-637, and Drp1 upregulation. De-phosphorylation of Drp1 at Ser-637 in particular induces calcium-calmodulin kinase (CaMKIIα) activity, which in turn increases GTPase signaling of Drp1 to increase mitochondrial fragmentation. Similarly, Slupe et al. [70] found that de-phosphorylation of Ser-656 of Drp1 in conditions of oxygen-glucose depravation increased Drp1 activity, leading to mitochondrial fragmentation, reduction of dendrite growth, and synaptic inhibition. Overall, Drp1 upregulation is caused by phosphorylation or de-phosphorylation of specific residues. The study of why Drp1 upregulation in diseases is caused by toxic proteins or respiratory conditions is thus worth further investigation, and may lead profound findings of how Drp1 may be specifically targeted for therapeutic treatment of diseases.

## 8. Mitochondrial Division Inhibitors and their Protective Effects

As discussed above, excessive mitochondrial fragmentation is the key factor in aging, and in several neurodegenerative diseases, inhibitors of mitochondrial fission may hold promise as therapeutic target. In the last two decades, there has been significant progress in identifying and developing several inhibitors of mitochondrial fission—these are as follows: Mdivi [71], P110 [72], Dynasore [73], DDQ [74], and mitochondrial division dynamin, Drp1 [75].

### 8.1. Mdivi 1

Several groups investigated the effect of reduced and/or partial inhibition of Drp1 in cell cultures [45,76] and mouse models [33,43,44,77] of AD (Figure 4).

#### 8.1.1. Genetic Studies

Reddy group [36] studied synaptic, dendritic, and mitochondrial proteins, and mitochondrial function, and GTPase enzymatic activity, in Drp1 heterozygote knockout mice. They found mitochondrial function and GTPase enzymatic activity appeared to be normal. However, free radicals and lipid peroxidation levels were significantly reduced in the Drp1 heterozygote knockout mice compared to the wild-type mice, indicating that partial Drp1 reduction boosts mitochondrial function and synaptic viability. Based on these observations they concluded partial reduction of Drp1 may have therapeutic implications to neurodegenerative diseases. 

Reddy group (Manczak et al. 2016) [77] studied whether a partial deficiency of Drp1 protects against Aβ-induced mitochondrial and synaptic toxicities in AD neurons [77]. They used genetic approach and crossed heterozygote knockout Drp1(+/−) mice with transgenic APP mice (Tg2576 strain) and created double mutant (APP × Drp1+/−) mice. Using cortical tissues from 6-month-old Drp1+/−, APP, APP × Drp1+/− and non-transgenic, control mice, they measured mRNA expressions and protein levels of genes related to the mitochondrial dynamics, mitochondrial biogenesis, and synapses. Using biochemical methods, they also studied mitochondrial function and measured soluble Aβ in brain tissues from all lines of mice. They found decreased mRNA expressions and protein levels of fission and matrix genes, and increased levels of fusion, biogenesis, and synaptic genes in 6-month-old APP × Drp1+/− mice relative to APP mice. They also found reduced mitochondrial dysfunction and reduced soluble Aβ levels were significantly reduced in APP × Drp1+/− mice relative to APP mice. Their observations suggest that a partial reduction of Drp1 reduces Aβ production, reduces mitochondrial dysfunction, and maintains mitochondrial dynamics, enhances mitochondrial biogenesis and synaptic activity in APP mice [77]. 

In another study, Reddy group (Kandimalla et al. 2016) [33] investigated whether a partial reduction of Drp1 protect neurons from phosphorylated Tau-induced mitochondrial and synaptic toxicities in AD progression [33]. They crossed Drp1+/− mice with transgenic Tau mice (P301L line) and created double mutant (Tau × Drp1+/−) mice. They studied mitochondrial dynamics, mitochondrial and synaptic genes in brain tissues from 6-month-old Drp1+/−, Tau, Tau × Drp1+/−, and non-transgenic control mice. Decreased mRNA and protein levels of fission and matrix and increased levels of fusion, mitochondrial biogenesis, and synaptic genes were found in 6-month-old double mutant mice relative to Tau mice. Mitochondrial dysfunction was reduced in Tau × Drp1+/− mice relative to Tau mice. Phosphorylated Tau found to be reduced in Tau × Drp1+/− mice relative to Tau mice. Based on these findings, they concluded that a partial reduction of Drp1 decreases the production of phosphorylated Tau, reduces mitochondrial dysfunction, and maintains mitochondrial dynamics, enhances mitochondrial biogenesis and synaptic activity in Tau mice [33].

#### 8.1.2. Pharmacological Studies

Baek et al. 2017 [43], noted that in their AD mouse model, upregulation of Drp1 was responsible for a host of mitochondrial dysfunctions, for which Mdivi-1 was used to inhibit. When treatment with Mdivi-1 was administered, the results were telling, as inhibition of Drp1 proved to be beneficial. Mdivi-1 inhibition of Drp1 reduced Aβ induced fragmentation of mitochondria, increased mitochondrial length, reduced ROS production, increased ATP production, reduced mitochondrial depolarization, and the affects aided the recovery of synaptic function [43]. The Drp1 inhibition alleviated Aβ-induced synaptic depression in hippocampal cells, whereas the untreated transgenic mice showed a marked ~50% decrease in exocytotic synaptic vesicles [43]. Remarkably, the Mdivi-1 also demonstrated a reduction of BACE1 gene transcription and translation, which was also evidenced by the reduction in Aβ peptides, oligomer formation, as well as aggregation into plaques [43]. Treatment with Mdivi-1 also improved creating new memory and cognition of transgenic mice, demonstrated by the ability to learn how to navigate hidden compartments and access targets, as well as evade punishment by means of a shock for an incorrect selection [43]. Equally important, the control mice demonstrated no adverse effects from their Mdivi-1 treatment. 

In 2017, Wang and colleagues [44] also studied the effect of Mdivi-1 on mitochondrial dynamics, particularly fission in CRND8 mice (transgenic APP strain). In Mdivi-1 treated and untreated 3-month-old CRND8 mice, they characterized mitochondrial fragmentation, distribution, and mitochondrial function in the pyramidal neurons. They found Mdivi-1 treatment rescued both mitochondrial fragmentation and distribution deficits and improved mitochondrial function in the CRND8 neurons both in vitro and in vivo. The mitochondrial dynamic deficits, Aβ1-42/Aβ1-40 ratio, and amyloid deposition were reduced by Mdivi-1 treatment. Further, cognitive deficits were prevented in Mdivi-1 treated CRND8 mice [44].

Additional research from Reddy’s lab [45] also concludes similar findings when experimenting with Mdivi-1 treatment of N2a cells expressing Aβ toxicity. The N2a cells treated with Mdivi-1 recovered biogenesis, and reduced oxidative damage, with lowered H_2_O_2_ production, as well as lowered lipid peroxidation. The mitochondria of the N2a cells also recovered cytochrome *c* oxidase activity as well, and reduced fragmentation [45]. Remarkably, the beneficial effects of Drp1 inhibition with Mdivi-1 were greater for N2a cells pre-treated with Mdivi-1, than treated with Mdivi-1 post Aβ42 toxicity [45].

Reddy group [78] was also capable of demonstrating Mdivi-1 usage as a way to mitigate Drp1 overexpression in Huntington’s disease. Considering that there is a correlation between mHtt accumulation and Drp1 translation, as well as Drp1 and mHtt interaction and HD progression, their team sought to find a correlation between the administration of Mdivi1 and the reduction of HD pathology [3,78]. Cells were harvested from Htt knock-in mice, and Mdivi1 treatments were either at 0, 25, or 50 µM, for 24 h, and harvested [60]. When Drp1 was inhibited by Mdivi-1, there was an increase in fusion genes and a decrease in fission genes transcribed, as well as an increase in cell viability [78]. In addition, there was a marked reduction in oxidative damage, measured by H_2_O_2_ production and lipid peroxidation, as well as increased biogenesis relative to ATP production [78]. 

In Parkinson’s disease, several groups extensively studied protective effects Mdivi-1. [79,80,81]. Bido et al. [80] studied the effect of Mdivi-1 in an overexpressed human A53T-α-synuclein (hA53T-α-syn) in the rat nigrostriatal pathway, with or without treatment of Mdivi-1. They found that Mdivi-1 reduced mitochondrial fragmentation, α-syn aggregation, neurodegeneration, and motor function. Mechanistically, Mdivi-1 reduced mitochondrial fragmentation, mitochondrial dysfunction, and oxidative stress. These in vivo results support the negative role of mutant α-syn in mitochondrial function and indicate that Mdivi-1 has a high therapeutic potential for PD. In another study, using two different mouse models of PD, Tieu group [81] investigated Drp1 inhibition using Mdivi-1. They found that blocking mitochondrial fission is neuroprotective in the PTEN-induced putative kinase-1 deletion (PINK1(−/−)) and 1-methyl-4-phenyl-1,2,3,6-tetrahydropyridine mouse models. They demonstrated that inhibition of the mitochondrial fission GTPase Drp1 using gene-based and small-molecule approaches attenuates neurotoxicity and restores pre-existing striatal dopamine release deficits in these animal models [81].

Short term studies such as those mentioned above bode well for small molecule therapeutics, though additional longitudinal clinical studies tracing long-term effects of Mdivi-1 administration would strengthen the premise of Mdivi-1 beneficence and ability to extend the lifespan of an organism post AD pathogenesis. 

In the aforementioned Down syndrome mouse model, Mdivi-1 was administered to mice to inhibit Drp1 overexpression. Valenti et al. [54] found that there was a considerable reduction in fragmentation, and improved mitochondrial dynamics, where there was a shift toward fusion, and longer, fuller shaped mitochondria, as well as increased intracellular ATP concentration [54]. In fact, the neuro progenitor cells harvested for analysis had improved survival, differentiation, and proliferation [54].

In summary, following the discovery of Mdivi-1 reported by Cassidy-Stone and colleagues in 2008 [71], over 236 papers (Pubmed search, August 10, 2019—https://www.ncbi.nlm.nih.gov/pubmed/?term=mdivi+1) have been published on Mdivi-1, noting that Mdivi-1 inhibits excessive mitochondrial fission and enhances mitochondrial fusion, and protects cells/neurons from mitochondrial dysfunction. Currently, Mdivi-1 is being considered for clinical trials for human diseases, including Alzheimer’s, Huntington’s, and Parkinson’s.

### 8.2. P110

P110, another Drp1 inhibitor, was used to reduce Drp1 overexpression in a Parkinson’s disease mouse model [82,83] and improved Drp1 related damage stemming from 1-methyl-4-phenyl-1,2,3,6-tetrahydropyridine (MPTP) toxin [82]. Normally, when MPTP oxidized to MPP^+^, intracellular Ca^2+^ is increased, inducing enzymes to propagate damage throughout cells, especially the lysosomes of astrocytes. Felichia et al. [82] used 9–10 week old, mice induced with MPTP toxicity, and P110 treatment was administered at 1.5 mg/kg/day for the best results. It was found that P110 prohibits Drp1 from being transported to the OMM of mitochondria, which prevents interaction with Fis1, and reduces fragmentation [82]. Apoptotic enzyme and pathway activities were reduced, and dopaminergic neurons demonstrated greater stability and survival when treated with P110 [81].

Guo et al. 2013 [84] investigated whether inhibition of excessive mitochondrial fission prevents mutant huntingtin-induced pathology. They used Drp1 inhibitor P110, and studied cell cultures of HD. They found that P110-TAT inhibited mtHtt-induced excessive mitochondrial fragmentation, improved mitochondrial function, and increased cell viability in HD cell culture models. P110-TAT treatment of fibroblasts from patients with HD and patients with HD with iPS cell-derived neurons reduced mitochondrial fragmentation and corrected mitochondrial dysfunction. Further, treatment of HD transgenic mice with P110-TAT reduced mitochondrial dysfunction, motor deficits, neuropathology, and mortality. They conclude that inhibition of DRP1-dependent excessive mitochondrial fission with a P110-TAT-like inhibitor may prevent or slow the progression of HD.

Using mouse models expressing demyelination congruent with human MS, a control group was compared with a set of mice treated with P110 [53]. Spinal cords of mice were harvested at 16 days, and then at 29 days, and Western blot analysis as well as immunochemistry were used to identify and quantitate proteins for analysis [53]. Not only was mitochondrial fission reduced after inhibition of Drp1, but also demonstrated a higher ratio of healthy oligodendrocytes [53]. 

Joshi et al. [49] noted similar effects when administering P110 to ALS model mice expressing G93A SOD1 mutation, where motor control continued to be enhanced during their longer 90-day study. P110 administration was shown to prevent the interaction between Drp1 and Fis1, as well as Drp1 translocation to mitochondria, which reduced mitochondrial fragmentation, and inhibited ALS pathology [49].

### 8.3. Dynasore

Dynasore is a dynamin GTPase inhibitor that is known to prevent the division or formation of dynamin-dependent endocytic vesicles. In 2006, Macia et al. [73] screened 16,000 small molecules to identify fission inhibitors and identified Dynasore that inhibits mitochondrial fission. They found that Dynasore interferes in vitro with the GTPase activity of Dynamin 1, Dynamin 2, and Drp1. There are small number of published studies that investigated the protective effects of Dynasore against excessive mitochondrial fragmentation and mitochondrial dysfunction [73,85,86] 

In 2013, Gao et al. [86] studied the protective effects of Dynasore against ischemia/reperfusion injury in mice. Langendorff-perfused mouse hearts were subjected to ischemia/reperfusion (30 min of global ischemia followed by 1 h of reperfusion). Pretreatment with 1 μM Dynasore prevented ischemia/reperfusion-induced elevation of diastolic pressure in the left ventricular end, indicating a significant and specific lusitropic effect for Dynasore. Interestingly, Dynasore treated oxidative stress-induced cultured adult mouse cardiomyocytes showed increased cardiomyocyte survival and viability, and reduced the depletion of cellular ATP. Moreover, the pretreatment of cultured cells with Dynasore protected mitochondrial fission induced by oxidative stress. Overall, Dynasore protected against cardiac lusitropy and limited cell damage through a mechanism that maintained mitochondrial morphology and intracellular ATP in the cells.

### 8.4. DDQ

DDQ (Diethyl (3,4-dihydroxyphenethylamino) (quinolin-4-yl) methylphosphonate) is a compound specifically designed to inhibit the levels of Drp1 and abnormal interaction of mutant proteins such as Aβ, with Drp1 [74]. In a study with human neuroblastoma (SHSY5Y+Aβ) cells that exhibited AD pathology, application of DDQ was able to disrupt Aβ’s interaction with Drp1. In the Aβ treated SHSY5Y cells, DDQ was able to reduce Drp1 and Aβ42 levels. Overall, mRNA and protein levels of mitochondrial dynamics (Drp1, Fis1, Mfn1, Mfn2), mitochondrial biogenesis (PGC1x, Nrf1, Nrf2, and TFAM), and synaptic genes (PSD95, synaptophysin, synapsin 1, synapsin 2, synaptobrevins 1 and 2 and GAP 43) were used to determine the therapeutic nature of DDQ. DDQ was able to enhance mitochondrial fusion, dynamics, synaptic proteins, and reduce mitochondrial fission proteins. These observations indicate that DDQ is a promising molecule that reduces excessive mitochondrial fragmentation and protect neurons mutant protein(s)-induced toxicities in mitochondrial diseases [74]. 

Overall, these studies suggest that Mdivi, Dynasore, P110, and DDQ are promising candidates to reduce excessive mitochondrial fission, to increase mitochondrial fusion, and to maintain mitochondrial function in cells/neurons in a large number of diseases that are affected by excessive mitochondrial fragmentation.

## 9. Is Mitochondrial Division Inhibitor 1 a Drp1 Inhibitor?

Mdivi 1 was also found to inhibit GTPase Drp1 activity by blocking the self-assembly of Drp1, resulting in the irreversible formation of elongated and tubular mt in wild-type cells [75]. As mentioned above, using molecular, cell biology, biochemical methods, and transmission electron microscopy methods, Reddy group studied preventive (Mdivi-1 + Aβ42) and curative (Aβ42 + Mdivi-1) effects against Aβ42 in N2a cells. Aβ42 was found to impair mitochondrial dynamics, lower mt biogenesis, lower synaptic activity, and lower mitochondrial function. On the contrary, Mdivi-1 enhanced mitochondrial fusion activity, lowered fission machinery, and increased biogenesis and synaptic proteins. Mitochondrial function and cell viability were elevated in Mdivi-1-treated cells [45]. 

However, Bordt and colleagues [87] questioned whether Mdivi-1 has any effect on mitochondrial fission, GTPase Drp1 activity, and mitochondrial elongation [87]. They argue that Mdivi-1 reversibly inhibits respiration at complex I and that the effects of Mdivi-1 on respiration and ROS are independent of Drp1. On a critical examination of the Bordt et al. study, it was clear that the entire study was circling around Sea Horse Bioanalyzer findings of Mdivi-1 treatment and complex I; they did not perform (1) mRNA and protein levels of mitochondrial dynamics, mitochondrial biogenesis, electron transport chain genes—complex I, III, IV, and V, (2) careful biochemical assays of ETC enzymatic activities, and (3) mitochondrial functional assays. Findings of their study are inconclusive, if one considers Mdivi-1 and/or any other mitochondrial molecules for treatment of human diseases with mitochondrial dysfunction. 

On the other hand, Reddy group (Manczak et al. [88]) conducted extensive cell biology, molecular biology, protein chemistry experiments, and ETC I, II, III, and IV enzymatic activities and Drp1-GTPase activity. They used (1) N2a cells treated with Mdivi-1, (2) overexpressed with full-length Drp1+Mdivi-1-treated N2a cells, and (3) Drp1 RNA silenced + Mdivi-1-treated N2a cells and studied: (1) mRNA and protein levels of ETC genes, mitochondrial dynamics, and mitochondrial biogenesis in untreated neurons and neurons treated with Mdivi-1, (2) enzymatic activities of complexes I, II, III, IV, (3) the mitochondrial network, (4) mitochondrial morphology, including size and number, (5) the extent of GTPase Drp1 enzymatic activity, and (6) the degree of mitochondrial respiration, using a Seahorse Bioanalyzer. They found reduced levels of the fission genes Drp1 and Fis1; increased levels of the fusion genes Mfn1, Mfn2, and Opa1; and the biogenesis genes PGC1α, Nrf1, Nrf2, and TFAM. Increased levels of complex I and IV genes were found. Immunoblotting data agreed with mRNA changes. Transmission electron microscopy analysis revealed reduced numbers of mitochondria and increased length of mitochondria in N2a cells treated with Mdivi-1, and in N2a cells overexpressed with full-length Drp1 and also treated with Mdivi-1. The mitochondrial network was increased. Increased levels of complexes I, II, and IV enzymatic activities were found in all three groups of cells treated with low concentration of Mdivi-1. Mitochondrial function was increased and GTPase Drp1 activity was decreased in all three groups of N2a cells. These observations strongly suggest that Mdivi-1 is a Drp1 inhibitor and directly reduces mitochondrial fragmentation. Based on these observations, they propose that Mdivi-1 is a promising molecule to treat human diseases with abnormalities in ETC complexes, I, II, and IV [77].

In preliminary preclinical studies of Mdivi-1, as discussed above, two groups independently treated mutant APP mice (APP/PS1 mice by Baek et al. 2017 [43]) and CRND8 mice (by Wang et al. 2017 [44]) with Mdivi-1 and studied beneficial effects of Mdivi-1. Preliminary results revealed that Mdivi-1 inhibition of Drp1 reduced Aβ-induced fragmentation of mitochondria, increased mitochondrial length, reduced ROS production, increased ATP production, reduced mitochondrial depolarization, and enhanced synaptic function. Findings from in vitro studies from our lab [87] and in vivo studies (in APP/PS1 and CRND8 mice) by Baek et al. [43] and Wang et al. [44] indicate that Mdivi-1 is a Drp1 inhibitor and directly reduces mitochondrial fragmentation and improves overall mitochondrial and synaptic functions in AD. 

However, further research is still needed on Mdivi-1, in order to determine whether Mdivi-1 affects electron transport chain complexes I, II, III, and IV from birth to terminal stages of disease progression in mouse models of AD. Since excessive fragmentation of mitochondria is the primary event in a several other neurological diseases, such as Huntington’s, Parkinson’s, ALS, Diabetes/Obesity, Down Syndrome, and multiple sclerosis, it is important to study Mdivi-1 effects in these mouse models as well. Currently, Mdivi-1 is being considered for clinical trials, and it is important to have comprehensive biochemical, molecular, cell biology, and electron microscopy data on Mdivi-1 in cell and rodent models.

## 10. Conclusions and Future Directions

Drp1 is critical for mitochondrial division and is essential for the distribution of mitochondria in axons, dendrites, and synapses. Drp1 is critical for mitochondrial division and is essential for the distribution of mitochondria in axons, dendrites, and synapses. Loss and/or overexpression dominant-negative mutation of Drp1 is involved in increased mitochondrial fusion and mitochondrial connectivity. 

Interestingly, several groups have found increased levels of Drp1 in postmortem brains and brain tissues from neurological diseases such as AD, HD, MS, PD, diabetes/obesity. These increased Drp1 levels in turn induce excessive fragmentation of mitochondria, reduced mitochondria fusion, increased free radical production, and defective mitochondrial function. Since mitochondrial fission has been found to be the major cellular event, inhibitors of mitochondrial fission may hold promise as a therapeutic target to treat patients diagnosed with such neurodegenerative diseases as AD and HD.

In the past 10 years, there has been some progress in identifying and developing inhibitors of Drp1 (or excessive mitochondrial fragmentation), including Mdivi-1, P110, Dynasore, and DDQ. Preliminary findings of these Drp1 inhibitors in in vitro (cell cultures) and in vivo (mouse) studies, provided beneficial effects, meaning reduced excessive mitochondrial fragmentation, enhanced mitochondrial fusion, mitochondrial network, and improved overall mitochondrial and synaptic functions. Thus, inhibition of Drp1 activity is often proposed for therapeutic mitigation of disease. Optimistically, Drp1 inhibition will be administered as a tool to treat mitochondrial dysfunction in neurodegenerative disease, but other diseases within the metabolic disease spectrum, as well as damaging effects of the aging process.

The development of additional Drp1 inhibitors for administration to additional animal models of AD and other neurological diseases will create the impetus to progress towards clinical trials where they will be most at use. Therefore, further research is still needed to test existing molecules from birth to terminal stages of disease progression and pathogenesis in different animal models. Further research is still needed to identify new Drp1 inhibitors.

## Figures and Tables

**Figure 1 cells-08-00961-f001:**
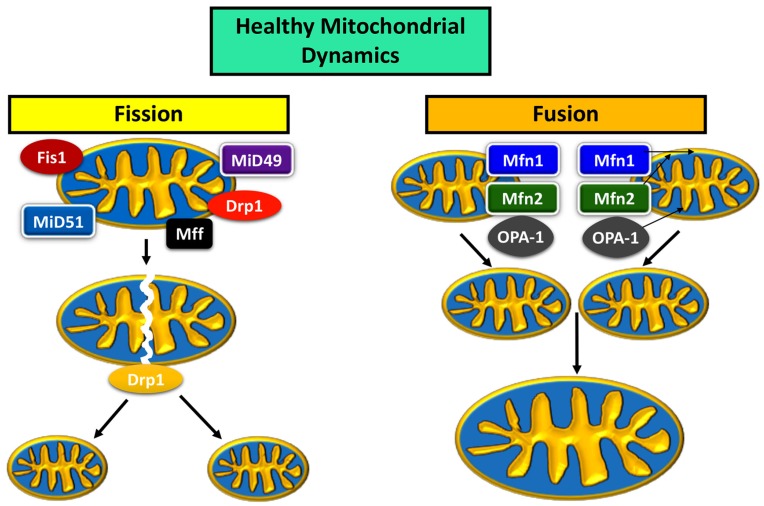
Mitochondrial dynamics in a healthy neuron.

**Figure 2 cells-08-00961-f002:**
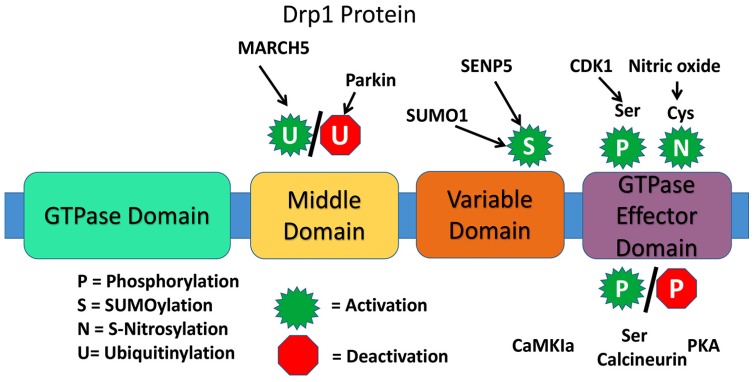
Structure of dynamin-related protein 1.

**Figure 3 cells-08-00961-f003:**
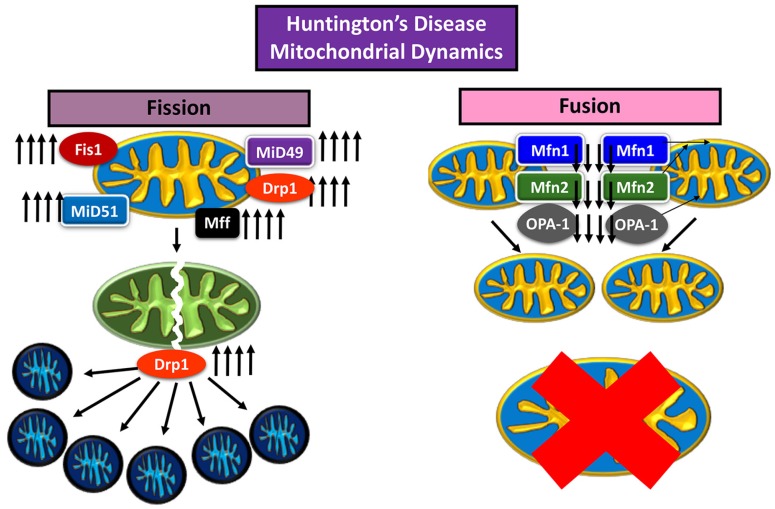
Impaired mitochondrial dynamics in Huntington’s disease neuron.

**Figure 4 cells-08-00961-f004:**
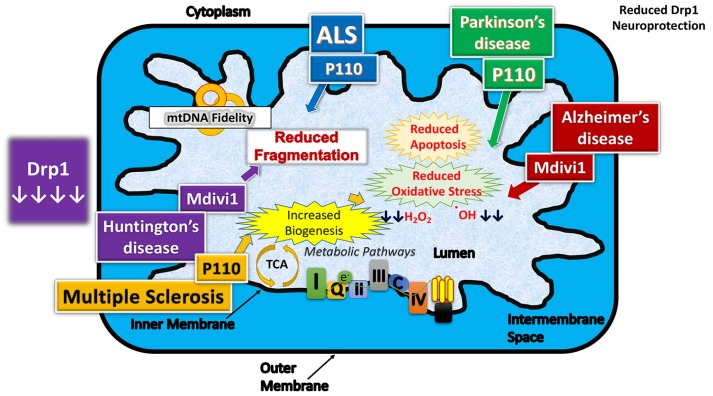
Therapeutic effects of reduced dynamin-related protein 1 in neurodegenerative diseases.

**Table 1 cells-08-00961-t001:** Dynamin related protein 1 and its phosphorylation sites.

Cell Type	Regulator Enzyme	Phosphoryl Group Addition Site	Effect	Citation
Mouse cortical neurons	Cdk5-mediated Drp1 phosphorylation at Ser^579^ is required for Aβ1-42 induced mitochondrial fission—GTPase domain activity is increased	Cdk5 could phosphorylate the recombinant Drp1 at Ser^579^. Aβ42 stimulation increased the phosphorylation level of Drp1. Cdk5 inhibitor roscovitine and knockdown of Cdk5 efficiently prevented Aβ42 induced Drp1 phosphorylation in neurons	Drp1 phosphorylation is activated by Ser^579^	Guo et al. (2018) [61]
HeLa (cervical cancer cells derived from Henrietta Lacks) Cyclin-dependent kinase 1 (Cdk1)	Ser^385^ (splice variant 1 of rat Drp1 Ser^616^), does not affect GTPase domain activity	Drp1 is activated during cell division when it accepts phosphoryl group from Cdk1	Drp1 is activated during cell division when it accepts phosphoryl group from Cdk1	Taguchi et al. (2007) [62]
HeLa	Protein kinase A (PKA)	Ser^637^ of Drp1’s GTPase effector domain	Drp1 GTPase domain activity is inhibited when phosphorylated by PKA	Chang and Blackstone (2007) [63]
HeLa neurons	Ca^2+^ calmodulin dependent PKI 1 (CaMKIα)	Ser^600^ (the Drp1 splice variant of Ser^637^)	Phosphorylation by CaMKIα activates Drp1	Han et al. (2008) [64]
Neonatal cardiomyocytes	Pim-1	Ser^637^	Drp1 phosphorylation by Pim-1 stops fission	Din et al. (2013) [65]
PC12	PKA, Calcineurin, and calcium coupling	Ser^656^ (splice variant 1 of rat Drp1 Ser^637^), does not affect GTPase domain activity	Phosphoryl group is transferred from PKA to Drp1, which stops GTPase activity, and inhibits apoptosis Calcineurin and calcium work to dephosphorylate and activate Drp1, which upregulates apoptosis	Cribbs and Strack (2007) [66]
	Calcineurin and calcium coupling	Ser^637^	Calcineurin and calcium work to dephosphorylate and activate Drp1, which upregulates apoptosis	Cereghetti et al. (2008) [67]

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
