# Peer review of "Dynamics of Dynamin-Related Protein 1 in Alzheimer’s Disease and Other Neurodegenerative Diseases"

_cells, 2019, doi:10.3390/cells8090961_

Round 1
Reviewer 1 Report
While the manuscript is detailed and offers a solid review of the impact that mitochondrial dynamics in neurodegenerative diseases, there are many issues that need to be addressed to help the readability of the manuscript:
1. The organization seems hastily conceived. There are sections and subsections that are not needed. The first to sections of the article “Introduction” and “Mitochondrial Dynamics in Homeostasis and Disease” could easily be combined as the contents of these sections are not so extensive to warrant their own section. Either add more details or combine to fit into a coherent module.
2. Similarly, the sections are ill defined and the contents of these sections do not always stick to the heading provided (i.e. Drp1 Isoforms has more to do with Drp1 Structure). And the references provided are often inadequate as the authors use the same citation over and over, even if that paper does not demonstrate the point.
3. The paper suffers from many run-on sentences that hurt the flow of the manuscript. At the end of the Homeostasis and Disease section, the last sentence has 71 words and 17 commas. This needs to be reworked and there are several places where the authors could be more concise.
4. And there are other places where additional comments are needed to distinguish whether increased mitochondrial fission in all neurodegenerative diseases has the same underlying mechanism or whether distinctions can be made. If they cannot, then why classify these things differently. Right now, many of the sections just read as a list of diseases that exhibit increased mitochondrial fission, but these could be combined to make a more efficient report unless there is a reason to highlight individual diseases. Based on this manuscript, I am not sure whether any distinctions exist.
5. The authors repeatedly claim that “mitochondrial division inhibitor 1 (Mdivi-1)” is “a Drp1 selective inhibitor”. This is largely agreed to not be the case. While it may selectively inhibit the yeast homolog of Drp1, Dnm1p, it does not directly impact Drp1. Rather, it indirectly impacts mitochondrial dynamics through inhibition of ETC function. Failure to acknowledge this seems reckless. The quest for a selective Drp1 inhibitor is ongoing. Some peptides (i.e. P110) may have a more direct impact on mitochondrial fission proteins, but mechanistic studies of mitochondrial fission inhbitors are largely lacking.
6. The authors missed an opportunity to discuss examples where mitochondrial fission is increased without cytotoxicity, which is observed in stem cell studies. I don’t know whether this is true to neural progenitors, but clearly not all mitochondrial fission is cytotoxic. If mitochondrial fission is to be targeted for neurodegenerative disease, is Drp1 the best target? And what are the potential complications with inhibiting a key cellular process like mitochondrial fission?
7. With the figures, the colors used are largely unwelcoming. Consider colors that can still be seen by color blind individuals. Specifically, yellow and green contrast is poor. Better use of contrasting colors will help.
8. For Figure 2 specifically, the overall organization is messy and the specific residues that are phosphorylated by the various kinases isn’t discussed until table 1 and much later in the text. Possibly consider rearranging the position of this figure and I mention below that a schematic of the primary sequence might be helpful. Also, Cdk5 isn’t mentioned.
9. In Table 1, PC12 calcium is listed as a regulatory enzyme, but it’s actually a mineral. It should not be listed here. And there are inconsistencies in which phosphorylations are activating and inhibitory. Is phosphorylation at S637 activating (CaMKIalpha activates Drp1)? If there are discrepancies, they need to be addressed.
10. What does Table 2 have to do with anything in the paper? The authors add another 40 citations at the very end with this table, but there is no reference to this in the manuscript. Very confused.
Additional comments for each section are provided below…
Introduction
Top of page 2: Use of the word ‘reproduction’ seems limited. To reproduce is to duplicate and make more of. Mitochondrial fission is important for partitioning mitochondria, but this process is also associated with apoptosis, mitophagy, changes in bioenergetic demands, etc. Mitochondrial division is more than just a binary fission event.
Mitochondrial Dynamics Homeostasis and Disease
There is only one sub-heading. It seems that a section on Disease is missing. And even the subsection on Homeostasis is very brief (5 sentences). Either expound on the topics presented or combine with previous section.
Mitochondrial Fusion and Fission
The authors state that there is “an ebb and flow between the expression of fusion and fission proteins”. I don’t know whether this is generally true for all cases. My impression is that the post-translational modifications are more dynamic in regulating the function of proteins that drive fission and fusion.
Last sentence of first paragraph on page 3… copy is misspelled as coy
Define the acronyms OMM and IMM as this is the first time they appear in the text
The “pinching” mechanism has been illustrated in EM papers, including Ingerman et al 2006 and Mears et al 2011 for yeast Dnm1p and Francy et al 2015 for human Drp1.
Dynamin-Related Protein 1
It is unclear how excessive mitochondrial fission can cause both toxicity/ cell death and extend lifespan in flies. Some additional discussion would be warranted.
Drp1 Structure
There is nothing about structure in this section. It seems like a throw-away. There are X-ray and cryo-EM structures of Drp1 and its partner proteins. There is an opportunity here to dicuss the domain architecture, but nothing. Again, I would either expound or merge this with another section.
Drp1 and its Isoforms
In this section, the authors get into the GTPase cycle as it related to structural mechanism. It is confusing and makes little sense to me. The one paper cited (ref 26) was work done with a fragment of the protein. It is hard to infer anything about the function of the protein from this fregment. I would look at other structural papers to learn more about this (Frolich et al. 2010?, Francy et al 2017, etc). Again, this seems like it is in the wrong section of the manuscript.
And again in the last paragraph of this section, it is unclear whether GTP binding is required for assembly on the mitochondrial surface. Since GTP hydrolysis is required for membrane constriction and this is followed by disassembly of the oligomers. I don’t think “assemblage” is a word and I don’t think peroxisomes are being disassembled… they’re undergoing division or fission similar to mitochondria.
Drp1 is not involved in “SUMOylation and phosphorylation and ubituitination”. It is subjected to these post-translational modifications, but it does not mediated these changes enzymatically, which is how I read it.
Drp1 and Abnormal Mitochondrial Dynamics
I would recommend that the sections that follow from this should be subsections. These diseases all exhibit abnormal mitochondrial dynamics, which is the major point of this section. Otherwise, this section offers little since it essentially introduces the diseases that will be summarized.
Drp1 and Alzheimer’s Disease
In the 1st paragraph of the section, ref 43 is cited 8 times1! Each sentence referes to the same citation, which seems odd. A summary sentence can be used with the reference and details provided for the experiments, similar to what was done in the next paragraph where ref 44 is only used once.
Drp1 and Amyotrophic Lateral Sclerosis
In this section, the authors state that “In ALS, just as with AD and HD, upregulation of Drp1 activity, as well as Fis1 activity, were at the center of mitochondrial fragmentation [49].” How was Drp1 activity assessed? And Fis1 does not have activity since it is not an enzyme. Was it overexpressed or something else? Additional clarity is needed?
Drp1 and Parkinson’s Disease
The authors claim “PINK1 also serves a role of activating Drp1 for scission necessary for mitophagy, by inducing the dissociation of protein kinase A (PKA) from A-kinase Anchoring Protein (AKAP1), allowing the phosphorylation of Drp1 at the OMM [29].” But PKA is an inactivating kinase for Drp1. Phosphorylation of Drp1 at S637 inhibits Drp1 recruitment to mitochondria, so loss of PKA at the surface of mitochondria would not allow for phosphorylation of Drp1 at the OMM, right? Confused.
Drp1 and Multiple Sclerosis and Drp1 and Down Syndrome
Both of these sections essentially refer to one paper each. And the same pattern of excessive fission is observed in both diseases. How is this different than other sections. It seems like the authors could just combine these all into one statement that exceesive mitochondrial fission is observed in several diseases , including …. and add a citation for each. If there is something unique about the way that this change in mitochondrial dynamics manifests, then highlight this. Otherwise, these very brief sections offer little on their own.
Drp1 and Diabetes/Obesity
The authors mentioned “increased Drp1 and GSK 3β levels altered mitochondrial morphology, reduced ATP production, and impaired activity of complex I.” But could it not also be argued that altered bioenergetics and ETC function in the db/db animals altered mitochondrial dynamics? Especially since complex I inhibitors, like Mdivi-1, can lead to altered mitochondrial dynamics.
Mechanism of Drp1 Dysfunction via Phosphorylation
Is it really Drp1 dysfunction? Or does phosphorylation regulate Drp1 function?
Also, Drp1 is phosphorylated at T595 by LRRK2 (Su and Qi, HMG, 2013).
Cyclin is misspelled and they mentioned that Drp1 is phosphorylated at Ser-579, which is the same as Ser 616, which was mentioned earlier. For the sake of simplicity and to avoid confusion, the authors should stick to one numbering scheme based on a specific Drp1 isoform. A schematic of the primary sequence with phosphorylation sites and kinases might be helpful here as a complement to the table.
Benefits of Partial reduction of Drp1 in Alzheimer’s and other neurological diseases
Much of this section is dedicated to Mdivi1 treatments in disease models. I think this section could be renamed “Benefits of reducing mitochondrial fission in neurological diseases”. I don’t know why AD is specifically highlighted and Mdivi can alter mitochondrial dynamics, but not through direct Drp1 interactions.
In Figure 4, it appears that P110 and Mdivi1 are acting inside the mitochondria, but I don’t think that is true for P110. It largely acts in the cytoplasm I thought. Mdivi does impact CI in the ETC, but this is not shown in the schematic. Again, the mechanisms of inhibition are confusing as presented.
Many of the examples in this section utilized Mdivi1, and the authors point out that long term studies would be useful to identify potential issues with Mdivi1. I thought that this small molecule has been shown to be fairly toxic, which would limit its impact.
Toxic Effects of Increased Drp1 in Aging and Neurodegenerative Diseases
This section seems largely redundant with previous sections. I am not sure what new information is presented here.
Mitochondrial division inhibitor 1, Reduced Drp1 Levels and Current Debate
I am glad that the authors acknowledge the debate, and they clearly fall on one side of this argument. They have tried to address this debate in their own work, so I think this section is particularly important.
It is not clear to me why there is a disparity in the CI inhibition that others have observed and that they do not see in their own work (Bordt et al). Is this cell type specific or why are they not seeing the same ETC defects?
Also, in this work, the authors to not directly examine Drp1 GTPase activity. With recombinant Drp1 protein, there is no impact observed with Mdivi1. So changes in the cellular Drp1 “activity” may reflect any number of changes as a consequence of alter bioenergetics (i.e. PTM changes, altered Drp1 interactions at the mito OMM, etc).
The authors mention… “Mdivi-1 is a promising molecule to treat human diseases with ETC complexes, I, II and IV [77]”. I am not sure what they are stating here. Diseases with ETC defects might respond to Mdivi1 treatment? Or something else.
Author Response
Reviewer 1
While the manuscript is detailed and offers a solid review of the impact that mitochondrial dynamics in neurodegenerative diseases, there are many issues that need to be addressed to help the readability of the manuscript:
1.The organization seems hastily conceived. There are sections and subsections that are not needed. The first to sections of the article “Introduction” and “Mitochondrial Dynamics in Homeostasis and Disease” could easily be combined as the contents of these sections are not so extensive to warrant their own section. Either add more details or combine to fit into a coherent module.
Response: as suggested, we combined “Mitochondrial Dynamics in Homeostasis and Disease” in the introduction section.
Similarly, the sections are ill defined and the contents of these sections do not always stick to the heading provided (i.e. Drp1 Isoforms has more to do with Drp1 Structure). And the references provided are often inadequate as the authors use the same citation over and over, even if that paper does not demonstrate the point.
Response: We sincerely appreciate reviewer’s comment. We modified this section, keeping general reader in mind. We added more relevance references and our revised version reads much better.
The paper suffers from many run-on sentences that hurt the flow of the manuscript. At the end of the Homeostasis and Disease section, the last sentence has 71 words and 17 commas. This needs to be reworked and there are several places where the authors could be more concise.
Response: As suggested, we modified long sentences not only at Homeostasis and Disease section (in the introduction of page 3) but also throughout the manuscript.
And there are other places where additional comments are needed to distinguish whether increased mitochondrial fission in all neurodegenerative diseases has the same underlying mechanism or whether distinctions can be made. If they cannot, then why classify these things differently. Right now, many of the sections just read as a list of diseases that exhibit increased mitochondrial fission, but these could be combined to make a more efficient report unless there is a reason to highlight individual diseases. Based on this manuscript, I am not sure whether any distinctions exist.
Response: we clarified mitochondrial dynamics (fission-fusion balance) in different neurodegenerative diseases and our revised version reads much better (see Page 16 of revised article). Since mitochondrial fragmentation is an important topic in neurodegenerative diseases, it is essential to keep the current status of mitochondrial dynamics in each disease separately.
The authors repeatedly claim that “mitochondrial division inhibitor 1 (Mdivi-1)” is “a Drp1 selective inhibitor”. This is largely agreed to not be the case. While it may selectively inhibit the yeast homolog of Drp1, Dnm1p, it does not directly impact Drp1. Rather, it indirectly impacts mitochondrial dynamics through inhibition of ETC function. Failure to acknowledge this seems reckless. The quest for a selective Drp1 inhibitor is ongoing. Some peptides (i.e. P110) may have a more direct impact on mitochondrial fission proteins, but mechanistic studies of mitochondrial fission inhibitors are largely lacking.
Response: We sincerely appreciate reviewer’s comments about Drp1 inhibitors – we now added a separate section on Drp1 inhibitors, covering Mdivi1, P110, Dynosore and DDQ on pages 15-22. We did our best based on current knowledge and published papers on Alzheimer’s and neurodegenerative diseases (FOCUS OF OUR ARTICLE).
We are not disagreeing with reviewer 1’s sentiment “mitochondrial division inhibitor 1 (Mdivi-1)” is “a Drp1 selective inhibitor”. This is largely agreed to not be the case. Mdivi 1 is tested in our lab extensively and others – we summarized the findings in our article. Again, as mentioned above, we summarized Drp1 inhibitors.
The authors missed an opportunity to discuss examples where mitochondrial fission is increased without cytotoxicity, which is observed in stem cell studies. I don’t know whether this is true to neural progenitors, but clearly not all mitochondrial fission is cytotoxic. If mitochondrial fission is to be targeted for neurodegenerative disease, is Drp1 the best target? And what are the potential complications with inhibiting a key cellular process like mitochondrial fission?
Response: We removed stem cells ‘Table 2’ and relevant references because it is beyond the scope of this article. Currently, we are writing another article that dedicated to mitochondrial dynamics in stem cells. We reconciled contradictions based on existing data – We all agree that we need more research on Drp1 and mitochondrial division inhibitor 1. We do not want take any sides and criticize any one here. Further research will provide answers for ‘mitochondrial fission is increased without cytotoxicity’. As per current knowledge and published papers on AD and other neurodegenerative diseases – Drp1 is increased and this increased Drp1 induces excessive fragmentation and mitochondrial dysfunction. As we know, the focus of our article is on Alzheimer’s and other neurodegenerative diseases – we balanced and did justice to our article.
It may be true that not all mitochondrial fission is cytotoxic in other human diseases, but not in Alzheimer’s and other neurodegenerative diseases.
With the figures, the colors used are largely unwelcoming. Consider colors that can still be seen by color blind individuals. Specifically, yellow and green contrast is poor. Better use of contrasting colors will help.
Response: we adjusted colors as much as we can and illustrate our message across. Mitochondrial readers can understand what we are trying to convey in the form of figures.
For Figure 2 specifically, the overall organization is messy and the specific residues that are phosphorylated by the various kinases isn’t discussed until table 1 and much later in the text. Possibly consider rearranging the position of this figure and I mention below that a schematic of the primary sequence might be helpful. Also, Cdk5 isn’t mentioned.
Response: We modified the figure, covering all 4 domains with changes, our modified figure is clear and good.
In Table 1, PC12 calcium is listed as a regulatory enzyme, but it’s actually a mineral. It should not be listed here. And there are inconsistencies in which phosphorylations are activating and inhibitory. Is phosphorylation at S637 activating (CaMKIalpha activates Drp1)? If there are discrepancies, they need to be addressed.
Response: We sincerely apologize for listing calcium as regulatory enzyme – we now modified as ‘calcium coupling’ in Table 1. We addressed the issue of phosphorylation at S637 activating (CaMKIalpha activates Drp1) in the text on pages 13-14. We also added Guo et al’s work of CDk5 and Ser 579 ref #61.
Currently, Drp1 phosphorylation/dephosphorylation is highly debated topic and we did our best in our article (not taking any sides).
What does Table 2 have to do with anything in the paper? The authors add another 40 citations at the very end with this table, but there is no reference to this in the manuscript. Very confused.
Response: We removed Table 2 and all related references. As mentioned above, we are writing a separate article stem cells and mitochondrial dynamics.
Additional comments for each section are provided below…
Response: We sincerely appreciate reviewer’s concerns – we carefully addressed reviewer 1's concerns in our revised manuscript.

Reviewer 2 Report
In this review the authors discuss the role of dynamin-related protein 1 (Drp1) in mechanisms of neurodegenerative diseases. Although the review is extensive and addresses the findings related to excessive mitochondrial fission and its consequent pathologic effect in individual neurologic diseases, only phosphorylation as post-translational modification that modulate the activity of Drp1 is discussed.
There are several conflicting data in the literature that show opposite functional effect of Drp1 phosphorylation of same serine residue under specific pathologic conditions.
The possible interactions between different post-translational modifications of Drp1 are not considered.
Similarly, only the Mdivi-1 inhibitor of Drp1 is mentioned in the review.
The individual paragraphs are missing conclusions and suggestions for further studies that are needed to advance the field.
Additionally, statements that would highlight the unresolved questions should be added to each discussed problem/disease .
Author Response
Reviewer 2
In this review the authors discuss the role of dynamin-related protein 1 (Drp1) in mechanisms of neurodegenerative diseases. Although the review is extensive and addresses the findings related to excessive mitochondrial fission and its consequent pathologic effect in individual neurologic diseases, only phosphorylation as post-translational modification that modulate the activity of Drp1 is discussed.
There are several conflicting data in the literature that show opposite functional effect of Drp1 phosphorylation of same serine residue under specific pathologic conditions. The possible interactions between different post-translational modifications of Drp1 are not considered.
Response: We sincerely appreciate reviewer 2’ comments – we extensively modified and addressed the issue ‘conflicting data in the literature that show opposite functional effect of Drp1 phosphorylation’. We addressed the issue of phosphorylation of Drp1 at S637 activating (CaMKIalpha activates Drp1) in the text, Figure 1 and Table 1 on pages 6-7 and 13-14. We also added Guo et al’s work of CDk5 and Ser 579 ref #61. Currently, Drp1 phosphorylation/dephosphorylation is highly debated topic and we did our best in our article (not taking any sides).
Similarly, only the Mdivi-1 inhibitor of Drp1 is mentioned in the review.
Response: We sincerely appreciate reviewer’s comments about Drp1 inhibitors – we now added a separate section on Drp1 inhibitors, covering Mdivi1, P110, Dynasore and DDQ on pages 15-22. We did our best based on current knowledge and published papers on Alzheimer’s and neurodegenerative diseases (FOCUS OF OUR ARTICLE).
The individual paragraphs are missing conclusions and suggestions for further studies that are needed to advance the field.
Response: We carefully checked and modified the text in each section and our revised version reads much better.
Additionally, statements that would highlight the unresolved questions should be added to each discussed problem/disease.
Response: we addressed this issue throughout the article and more clearly in conclusions and future directions section on pages 24-25. We extensively covered mitochondrial division inhibitors on pages 15-22.

Reviewer 3 Report
This is a comprehensive review of the state-of-the-art of Drp-1 in the mitochondria dynamics and its implication on the development of several neurodegenerative diseases. Moreover, the authors conclude that the inhibition of Drp-1 could be envisaged as a promising strategy to manage neurodegenerative disorders.
Author Response
We sincerely thank reviewer 1 for supporting our article.
Round 2
Reviewer 2 Report
The authors addressed all the questions pointed out by the reviewer . I have no more comments.